# Hydrolats from *Humulus lupulus* and Their Potential Activity as an Organic Control for *Varroa destructor*

**DOI:** 10.3390/plants11233329

**Published:** 2022-12-01

**Authors:** Azucena Elizabeth Iglesias, Giselle Fuentes, Giulia Mitton, Facundo Ramos, Constanza Brasesco, Rosa Manzo, Dalila Orallo, Liesel Gende, Martin Eguaras, Cristina Ramirez, Alejandra Fanovich, Matias Maggi

**Affiliations:** 1Centro de Investigación en Abejas Sociales (CIAS), Facultad de Ciencias Exactas y Naturales, Universidad Nacional de Mar del Plata (UNMdP), Mar del Plata CP 7600, Argentina; 2Instituto de Investigaciones en Producción, Sanidad y Ambiente (IIPROSAM), Consejo Nacional de Investigaciones Científicas y Técnicas (CONICET), Universidad Nacional de Mar del Plata (UNMdP), Mar del Plata CP 7600, Argentina; 3Laboratorio de Investigaciones en Evolución y Biodiversidad, Universidad Nacional de la Patagonia San Juan Bosco, Esquel CP 9200, Argentina; 4Departamento de Química y Bioquímica, Facultad de Ciencias Exactas y Naturales, Universidad Nacional de Mar del Plata (UNMdP), Mar del Plata CP 7600, Argentina; 5Instituto de Investigaciones en Ciencia y Tecnología de Materiales (INTEMA), Consejo Nacional de Investigaciones Científicas y Técnicas (CONICET), Universidad Nacional de Mar del Plata (UNMdP), Mar del Plata CP 7600, Argentina

**Keywords:** volatile compounds, beta-caryophylene oxide, *Varroa destructor*, hydrolats, terpenoids, *Apis mellifera*

## Abstract

*Varroa destructor* is a parasitic mite, which is considered a severe pest for honey bees causing serious losses to beekeeping. Residual hydrolats from steam extraction of hop essential oils, generally considered as a waste product, were tested for their potential use as acaricides on *V. destructor*. Four hop varieties, namely Cascade, Spalt, Victoria, and Mapuche, showed an interesting performance as feasible products to be used in the beekeeping industry. Some volatile oxidized terpenoids were found in the hydrolats, mainly β-caryophyllene oxide, β-linalool, and isogeraniol. These compounds, together with the presence of polyphenols, flavonoids, and saponins, were probably responsible for the promissory LC_50_ values obtained for mites after hydrolat exposition. Victoria hydrolat was the most toxic for mites (LC_50_: 16.1 µL/mL), followed by Mapuche (LC_50_ value equal to 30.1 µL/mL), Spalt (LC_50_ value equal to 114.3 µL/mL), and finally Cascade (LC_50_: 117.9 µL/mL). Likewise, Spalt had the highest larval survival, followed by Victoria and Mapuche. Cascade was the variety with the highest larval mortality. In addition, none of the extracts showed mortality higher than 20% in adult bees. The Victoria hydrolat presented the best results, which makes it a good compound with the prospect of an acaricide treatment against *V. destructor.*

## 1. Introduction

*Apis mellifera* [1] is a bee species with a cosmopolitan distribution that can be found today on almost all continents. It is widely used in beekeeping and pollination services, as it is one of the main pollinating species for crops and wild plants [2,3].

In recent years, this species has faced a steep decline in its colonies due to several factors, among which the ectoparasite mite *Varroa destructor* [4] generates great conflicts in the beekeeping activity, as well as complications for all crops that depend on the pollination services provided by *A. mellifera* [5,6]. It is also important to note that part of the parasite-host imbalance observed between these two species is mainly due to the short history of association between *V. destructor* and honey bees, where the latter has not particularly developed regulatory mechanisms to counteract the harmful effects of this parasite [7]. Over the years, it has been demonstrated that if no control mite measures are taken, bee colonies die in one season [6].

Since the beginning of the pathology caused by *V. destructor* mites, beekeepers have used a wide variety of acaricides to control this parasite in colonies, which mainly are substances from chemical families such as organophosphates and pyrethroids [8,9,10]. However, over the years, it has been observed that the use of these acaricides can cause significant damage to bee populations [11,12,13], in addition to contaminating bee products [10,14,15] or even worse, impact on bee health [16,17]. Especially, the indiscriminate application of these substances led to the emergence of resistant populations of mites to these compounds [18,19,20,21,22,23,24,25].

As a consequence of the negative impact on bee colonies observed for synthetic acaricides, different researchers have studied and developed alternative forms of control [6,26,27]. Among them, organic acaricides have been tested, with characteristics that make them suitable for use in the environment [28], such as low toxicity to mammals and low environmental impact. These molecules are indispensable in the design of Integrated Pest Management programs, which combine techniques and knowledge that allow the rotation of synthetic acaricides. When these techniques are well-implemented, they allow sustainable development of the apiary with a lower environmental impact and high-quality products [29,30].

There are substances of natural origin with excellent results in the control of bee pests; among them, the essential oils (EOs) of various plant species have been extensively studied [31,32]. EOs are secondary metabolites that facilitate the main metabolism of plants and are widely used for defense against parasites [28,33]. In this regard, botanically derived products have demonstrated a wide range of biological activities, including toxicity, repellency, and growth regulatory properties [34,35,36,37].

In recent years, research has been conducted with various methods for the extraction of secondary metabolites from plant material of *Humulus lupulus* species, with promising results for the control of *V. destructor* [38,39,40,41,42,43,44]. The species *H. lupulus* L. belongs to the Cannabaceae family, and its female flowers (commonly known as hop cones) are traditionally used in the brewing industry to add flavor and bitterness to beer [45]. The main structural classes of chemical compounds identified in the essential oils of hop flowers are terpenes, bitter acids, and chalcones. Hops are also rich in flavonol glycosides (kaempferol, quercetin, quercitrin, rutin) [46] and catechins (catechin gallate, epicatechin gallate) [47]. Hundreds of terpenoid components have been identified in the volatile oil (0.3–1.0 wt.% of hop weight): mainly β-caryophyllene, farnesene, humulene (sesquiterpenes), and myrcene (monoterpene) [48,49,50].

Although there are numerous investigations focused on *H. lupulus* extractions in polar and non-polar solvents, to date, there are no studies that have evaluated the acaricidal activity of hydrolats of this species against *V. destructor*. Hydrolats, commonly known as flower water, is a by-product of obtaining essential oils by hydrodistillation, currently used in different industries such as food and cosmetics for their organoleptic and biological properties. The hydrolats of different plant species have been shown to have biological activity and are also used in agriculture against different pests such as fungi and insects and also for soil fertilization [51,52,53]. In general, there are not many studies that evaluate the role of the volatile compounds of hydrolats as potential acaricides. However, the main components are generally the same as those present in the oxygenated fraction of the corresponding essential oils [54]. 

The aim of the present research was to analyze the biological activity of hydrolats, obtained as water-soluble fractions in the hydro distillation process of essential oils from female flowers of the Spalt, Cascade, Victoria, and Mapuche varieties of *H. lupulus* over *V. destructor,* bee larvae, and adult bees, as well as to evaluate the attractive or repellent properties for these hydrolats on *V. destructor.*

## 2. Results

### 2.1. Chemical Composition

The volatile composition of the hydrolats is summarized in Table 1. The four varieties of hops presented a high presence of volatile compounds in their oxidized form, partly due to the extraction methodology.

The qualities of these extracts on the total content of phenolic compounds are shown in Table 2. For the Victoria variety, a higher amount of saponins and a higher antioxidant activity were found, followed by Cascade and Mapuche hydrolats with similar values between them. Finally, with respect to the Spalt hydrolat, a higher proportion of flavonoids and polyphenols in total were found, with lower antioxidant activity and a lower presence of saponins (Table 2).

### 2.2. Mite and Bee Lethality Test

The estimated LC_50_ value for *V. destructor* at 24 and 48 h obtained for each hydrolat is given in Table 3. The hydrolat from the Victoria hop variety demonstrated the lowest value of LC_50_ against *V. destructor*, indicating that it is the most lethal variety for mites. The observed nurse bee toxicity did not show significant mortality compared to the control. NOAEL values were calculated for each variety. For the Spalt hydrolat variety, the NOAEL was ≥20 μL/mL (X2(3, N = 25)) = 0 with a *p*-value of 1; for the Cascade hydrolat, the NOAEL was ≥20 μL/mL (X2(3, N = 25)) = 0 with a *p*-value of 1; for the Mapuche variety, the NOAEL value was of ≥20 μL/mL (X2(3, N = 25)) = 0.7003 with the *p*-value of 0.9512; and finally the NOAEL value calculated for the Victoria variety hydrolat was ≥20 μL/mL (X2(3, N = 25) = with a *p*-value of 0.9512. For tau-fluvalinate, the LC_50_ estimated at 24 h was 2.89 mL/dish for mites. This value decreased to 2.03 mL/dish at 48 h. LC_50_ for bees was 1.682 and 1.427 mL/dish at 24 h and 48 h, respectively. Significant differences were detected in relation to treatments (*p* < 0.001).

### 2.3. Attractivity Test

The hydrolats from all varieties tested did not show signs of attracting or repelling *V. destructor* mites. Fisher’s analysis showed that the expected position of the mites in the Petri dishes did not show significant differences with the control and was similar between the groups. The *p*-values obtained for the hydrolats were: Cascade variety (*p*-value = 0.5991), Spalt variety (*p*-value = 0.2489), Mapuche variety (*p*-value = 0.3795), and Victoria variety (*p*-value = 0.3215).

### 2.4. Larvae Lethality Test

On the last day of the larval stage, larval survival was different for each treatment, with significant differences between the larval survival curves (X2(5, N = 250) = 46,12, *p*-value = 0.0001). In particular, honey bee larvae exposed to the Mapuche and Cascade varieties showed significant differences in survival compared to the control group (Log-rank (Mantel--Cox) Test *p* = 0.0015; *p* < 0.0001, respectively). Victoria presented a larval survival on the sixth day of 80.39% ± 3.32, Cascade presented 69.10% ± 3.76, and Mapuche larval survival was 73.76% ± 3.58. Spalt showed a higher larval survival on the sixth day, 90.30% ± 2.55, and the control without solvent presented a larval survival of 85.09% ± 3.23. Finally, the control with solvent had a survival of 86.82% ± 3.02. The results can be seen in Figure 1.

When analyzing the weights of larvae on the last day of the experiment, no differences were found between the weights, except for the Spalt variety, which was higher than the control (Figure 2).

## 3. Discussion

It has become necessary to develop alternative control strategies against *V. destructor* to synthetic control substances that generate drawbacks such as residues and acaricide resistance [15]. The use of organic acids and compounds from plants are shown as interesting alternatives [55]. The four varieties of hops presented a high presence of volatile compounds in their oxidized form, partly due to the extraction methodology and the qualities of these extracts. The Victoria hydrolat presented a high acaricidal activity and null toxicity in adult bees. The larvae showed survival of almost 80.39% on the sixth day, which makes it a good compound to analyze its prospect as an acaricide treatment against *V*. *destructor.* The Victoria hydrolat presented the highest toxicity against mites, similar to those reported for other plant extracts [43]. For example, Zaitoon (2001) showed that acetonic extracts from *Rhazya stricta*, *Heliotropium bacciferum*, and *Azadirachta indica* had remarkable in vitro toxicity against *Varroa* mites [56]. At the same time, Damiani et al. (2011) evaluated the biological activity of botanical extracts from two indigenous plants from South America and obtained high levels of toxicity against the mites and no effect on *A. mellifera* [57]. The results obtained in the present study would indicate that Victoria hydrolat is a good compound to be used as an alternative for the control of *V. destructor*.

With regard to the other varieties studied in this work, a lower acaricidal activity was observed from the Spalt variety, which may be due to its low saponin content. According to Armah, et al. (1999), the spontaneous formation of a complex between the saponins and the membranes formed a micellar-type structure in two directions until the formation of the pore, thus allowing the passage of macromolecules into the cell [58]. Saponin compounds, together with the synergy caused by other components, have a great capacity to break the cells accelerating the cell death process, thus allowing cell death and collapse [59]. Another consideration is that the Spalt hydrolat had a lower presence in the number of compounds, as well as a lower concentration of them, as is the case of β-caryophyllene. The latter has been reported to have high toxicity at extremely low LD_50_ for other arthropods, such as dust mites, or one of the most important pests of various crops, the red spider mite, as it is commonly called [60]. In particular, β-caryophyllene oxide usually appears in different types of extractions in a lower proportion in essential oils [61], but in these types of hydrosols, this compound was found in high proportions for all cases. In the reports performed by Oh et al. (2014), β-caryophyllene was tested against two species of dust mites, with very favorable results, obtaining LD_50_ values of 0.44 µg/mL at 24 h after treatment [61]. In our study, the varieties that presented the highest percentage of this compound (Victoria and Mapuche) showed the best results against Varroa mites, with LC_50_ values of 16.1 µL/mL and 30.1 µL/mL, respectively (24 h after treatment). It must be considered that the LD_50_ value is always lower than the LC_50_ value (OECD). In the present study, high amounts of this compound were found in the hydrolat phase of all tested varieties.

Another compound that deserves attention is Linalool. This compound was mostly found in the four hydrolats analyzed in the present research, but also was found in the EO of many other plant varieties with interesting acaricidal properties for study [62,63]. Bava et al. (2021) found Linalool in all EOs studied, but in a very high concentration in bergamot EO, which consistently reduced *V. destructor* viability [64]. The authors point out that probably the combination of different concentrations of these compounds contributes to the successful mite inactivation.

No effects on adult bees were found for the four studied varieties. Regarding larval toxicity, the Cascade and Mapuche varieties showed the lowest survival values, with 69% survival for Cascade and 73% for Mapuche. This is relevant, taking into account that a good acaricide should reduce mite infestation without causing high toxicity and lethality in honey bees [65].

In many studies with essential oils, it is common to introduce the analysis of attraction and repellency on the mite since the result can give us important indications on the best use of these compounds [57]. Any effect that interferes with the mite’s ability to locate its host may have a practical value as a method of control. A substance able to modify the mite´s behavior inside the honey bee colony can be useful in controlling *V. destructor* [66]. In all cases, no statistically significant effect was observed with respect to the attraction or repellency of these compounds against *V. destructor*.

In recent years, studies on organic compounds with acaricidal activity, in general, and more particularly in the area of bee health, have increased [22,43,57,65,67]. However, we cannot ignore the fact that the honey bee is faced with various stressors in the different environments where it is found and under the different economic practices that are carried out with its use. Thus, the study of compounds of organic origin, and even more, as in this case of industry residues, is an important step towards the implementation of an ecologically and environmentally acceptable mechanism of *V. destructor* control in the industry.

## 4. Materials and Methods

### 4.1. Mites and Bees

*A. mellifera* workers and larvae, as well as adult females of *V. destructor*, were obtained from the experimental apiary of the Centro de Investigación de Abejas Sociales (CIAS of the Universidad Nacional de Mar del Plata, 38°10′06″ S, 57°38′10′10″ W) during the summer and fall of the 2018–2019 season. One year before the start of the experiments, the hives were treated with Aluen Cap^®^ (oxalic acid-based organic acaricide) [6]. Then, the hives were allowed to be re-infested by the mite naturally. Therefore, colonies with less than 1% of mite infestation in adult worker bees were considered healthy, and those colonies with greater than 5 % of mite infestation in adult bees were considered infested colonies with high levels of *V. destructor* to be used as the source of the mites for the trials. Combs with operculated brood from colonies with high infestation levels were selected and taken to the laboratory for subsequent collection of adult females of *V. destructor* from sealed cells, selecting those with dark brown color. Nurse bees were collected from healthy hives, 1–3 days old, from frames with open brood containing eggs and larvae up to larval stage L5-L6 (Iglesias et al., 2021). One-day-old larvae (L1) were collected from frames with open brood from healthy hives. Bees (adults and larvae) and mites were collected from at least 5 different *A. mellifera* hives from the apiary.

### 4.2. Plant Material

About 100 g of *H. lupulus* female flower cones of Mapuche, Cascade, Spalt, and Victoria varieties were collected in February 2018 from a lupulus cultivar located in the vicinity of Mar del Plata (38°10′06′′ S, 57°38′10′10″ W, Buenos Aires province, Argentina). Flowers were dried at 55 °C and stored once they reached 9% humidity. Then, a two-hour hydro distillation was performed using a Clevenger-type European Pharmacopoeia apparatus [68]. Once this extraction process was completed, the oils and the remaining hydrolat were separated and placed in caramel-colored glass jars and kept at 4 °C until testing. The identification codes of the varieties are Spalt: IIMyCher: MDQ: 00455; Cascade: IIMyCher: MDQ: 00456; Mapuche: IIMyCher: MDQ: 00458; and Victoria: IIMyCher: MDQ: 00460.

### 4.3. Volatile Compounds

#### GC-MS Analysis

Samples were analyzed using a Shimadzu GCMS-QP2100ULTRA-AOC20i with a column of 0.25 mm ID, 30 m, and 0.1 µm phase thickness Zebron ZB-5MS. Samples were injected at pulsed splitless mode, and the injection volume was 1 mL. The interface and the ionization source were kept at 300 °C and 230 °C, respectively. Helium chromatographic grade (99.9999%) was used as the carrier gas with a constant linear velocity of 52.1 cm/seg. The oven temperature program started at 50 °C, where it was held for 2 min, and then increased to 300 °C at 15 °C/min, where it was held for 4 min. Electron impact ionization (EI) was used at 70 eV in a full scan. Full-scan EI spectra were acquired under the following conditions: mass range 35–700 *m*/*z*, scan time 0.3 s, and solvent delay 3.0 min. Characterization was performed using NIST and Wiley libraries and retention index.

### 4.4. Total Content of Phenolic Compounds

The total content of phenolic compounds in the different extracts was determined by the Folin--Ciocalteu method (F-C), according to the procedure reported by Singleton and Rossi (1965), with some modifications [69]. The F-C assay is a reaction based on electron transfer that measures the reducing capacity of an antioxidant. The F-C reagent is a mixture of phosphotungstic acid (H_3_PW_12_O_40_) and phosphomolybdic acid (H_3_PMo_12_O_40_) that reacts with phenols and non-phenolic reducing substances to form chromogen. The latter can be detected spectrophotometrically since, under alkaline conditions, the oxotungstate and oxomolybdate formed in this redox reaction show a blue coloration proportional to the concentration of polyphenols [70]. To carry out the quantification of phenolic compounds, a calibration curve was performed from a standard solution of Gallic Acid (GA) of 200 µg/mL, obtaining the following concentrations: 0, 2, 6, 8, 10, 15 and 20 µg/mL. From the linear regression of the sample, the total polyphenol content of each extract was estimated. The determination of the total amount of polyphenols for each of the extracts was carried out using a 96-well microplate in quadruplicate and was calculated as mg gallic acid equivalents (GAE) per g of dry extract.

#### 4.4.1. Total Flavonoid Content

Total flavonoid content was determined by the method of Woisky and Salatino 1998, with some modifications [71]. The calibration curve was constructed from a quercetin (QE) standard solution of 200 µg/mL. The following concentrations were used; 0, 2, 4, 6, 10, 14, 18, and 22 µg/mL. From the linear regression of the sample, the total flavonoid content of each extract is obtained. The determinations were carried out in quadruplicate, and the absorbance was measured at 420 nm using an Agilent 8453 UV-visible spectrophotometer with a diode array. Total flavonoid content was calculated as mg QE equivalents per g of dry extract.

#### 4.4.2. Total Saponin Content

The determination of total saponin content (TSC) was performed using the methodology proposed by Le et al. (2018), with some modifications [72]. The principle of the method is based on the reaction of triterpene saponins, which after being oxidized by sulfuric acid, react with vanillin. This reaction causes a distinct change in coloration towards purplish red and can be measured at wavelengths ranging from 473 to 560 nm. The TSC of a plant sample is determined from a calibration curve with a standard saponin (e.g., escin, oleanolic acid, diosgenin, quillaja saponin) and expressed in terms of equivalence of the standard [72]. The calibration curve was constructed with oleanoic acid (OA) as a reference standard, and concentrations between 0.001–0.005 µg/mL of OA were obtained, with which the calibration curve was performed. From the linear regression of the calibration curve, the TSC of each extract was obtained. The absorbance at 560 nm was measured in an Agilent 8453 UV-visible spectrophotometer with a diode array, and the values were recorded. Treatments were performed in quadruplicate, and the TSC was calculated as mg OA equivalent per g of dry extract.

### 4.5. Mite and Bee Lethality Test

The bioactivity of *H. lupulus* hydrolats was determined using the complete exposure method [31]. This method was also used in other research studies to test the acaricide activity of natural substances [21,43,44,65,73,74,75] The hydrolats of 4 hop varieties (Cascade, Victoria, Spalt, and Mapuche) were diluted in acetone in different treatments with concentrations of 0 (solvent-control-only), 2.5, 5, 10, and 20 µL/mL. Control treatments consisted of Petri dishes filled with solvent, acetone as a negative control, and fluvalinate (Sigma Aldrich, Darmstadt, Germany) as a positive control. An aliquot of 1mL of each concentration and controls was placed in each Petri dish with a diameter of 10 cm, then allowed to evaporate in free air for 3–5 min. After this time, 5 nurse bees and 5 female *V. destructor* mites obtained from brood cells were placed in the dish. Petri dishes containing mites and bees were incubated at 30 °C with 70% humidity for 72 h. The bees were fed with 3 g of candy per plate. A total of 5 treatments were performed with 5 plates per group. Mite and bee mortality was recorded every 24, 48, and 72 h by direct observation using an ocular magnifying glass. With this data, the LC_50_ (lethal concentration 50) lethality index was obtained. Acetone was purchased from Anedra, Research AG (Tigre, Buenos Aires, Argentina).

### 4.6. Attractivity Test

In order to test the attraction or repellent effects against *V. destructor* for Cascade, Victoria, Spalt, and Mapuche hydrolats, the methodology proposed by Damiani et al. (2011) was used [57]. Each Petri dish (9 cm diameter) was divided into four sections (A, B, C and D). For the hydrolats treatments, a circle of filter paper (1 cm diameter) embedded with 8 μL of acetone was placed in section A, and in section D, a circle of filter paper (1 cm diameter) embedded with 8 μL of the LC_50_ calculated for each hydrolat variety. As control tests, a circle of uncontaminated filter paper was placed in section A, and another circle of filter paper embedded with 8 μL of acetone was placed in section D. The solutions were allowed to evaporate before testing. Next, a single adult female mite was placed in the center of the Petri dish. After 90 min, the location of the mite was recorded (as in A, B, C, or D). Thirty observations for each hydrolats and controls were run simultaneously.

### 4.7. Honeybee Larvae Lethality Test

To test the acute toxicity of hydrolats in bee larvae, a bioassay based on the previous work of Iglesias et al., 2020 was used [43]. One-day-old larvae (1st instar, L1) of *A. mellifera* were collected from combs of healthy (non-infested) colonies and used for in vitro experiments with 96-well culture plates. The following treatments were performed: (a) control without solvent, (b) control only with solvent (acetone), (c) Spalt hydrolats, (d) Cascade hydrolats, (e) Mapuche hydrolats, and (f) Victoria hydrolats. An aliquot of 1 μL of the LC_50_ for each hydrolat and control was placed in each well. This solvent was selected because of its low toxicity in bee larvae [76]. One culture plate per treatment was used. The aliquot was left to evaporate in each well before the larvae were transferred from the brood comb to plates, with a single larva added to each well. Fifty replicates per treatment were made. The plates were then placed into a desiccator maintained at a relative humidity of 96% (K_2_SO_4_ saturated) in a 34 °C incubator. The volume and composition of the diet provided daily to larvae followed the protocol by Aupinel et al., 2005 [77]. Larval mortality was recorded when an immobile larva, even under paintbrush contact, was observed, as it is considered dead. To evaluate if hydrolats impact larval weight, fresh weight was measured in individuals on day 7 [78].

The following components 2,2′-Azino-bis (3-ethylbenzothiazoline-6-sulphonic acid), diammonium salt (ABTS), and Folin--Ciocalteu reagent were purchased from Sigma-Aldrich^®^ (Darmstadt, Germany). The following standard compounds were purchased from Sigma-Aldrich^®^ United States, and all standards purity ≥ 98.0%: gallic acid, quercetin, Trolox, vanillin, and oleanoic acid. Acetone and sulfuric acid were purchased from RESEARCH AG (Anedra, Tigre, Buenos Aires, Argentina) and used as received.

### 4.8. Statistical Analysis

#### 4.8.1. Analysis of Mites and Bee Lethality Test

Mites and bee mortality were determined by visual inspection of the Petri dishes after 24, 48, and 72 h. The LC_50_ values for *V. destructor* and inverse 95% confidence intervals were estimated. The probit function and the PROC LOGISTICS procedure were used to transform variables and to resume the information [79,80]. The highest concentration of essential oil, which did not induce bee mortality significantly higher than that observed in controls (No Observed Adverse Effect Level = NOAEL (*p* = 0.05)), was estimated for bees [76]. The statistical comparison between uncorrected mortality in the treated sample and the control was performed using the Chi^2^ test. 

#### 4.8.2. Analysis of Attractivity

The attractivity of hops hydrolats to mites was analyzed by means of a binomial test for a two-level categorical dependent variable using the Fisher’s test and the software R Commander. R package version 2.5–3 (Fox and Bouchet-Valat, 2019).

#### 4.8.3. Analysis of Honeybee Larvae Lethality Tests and Weight

Survival and mortality of bee larvae were estimated using the Graph pad software, according to Iglesias et al. (2020) [43]. Analysis of larval weights were compared by full interaction ANOVA analysis using log-transformed data to linearize potential functions, given that the required assumptions were maintained.

## 5. Conclusions

Our study suggests the good in vitro efficacy of the Victoria hydrolat, which presented a high acaricidal activity, null toxicity in adult bees, and very low toxicity in larvae of honey bees. These results would indicate that Victoria hydrolat is a good compound to use as an alternative for the control of *V. destructor*. In conclusion, this study represents another example of potential eco-friendly bee pesticides, leaning research towards the discovery and use of natural preparations rather than synthetic molecules whose persistent use and accumulation in the environment have proven to be counterproductive.

## Figures and Tables

**Figure 1 plants-11-03329-f001:**
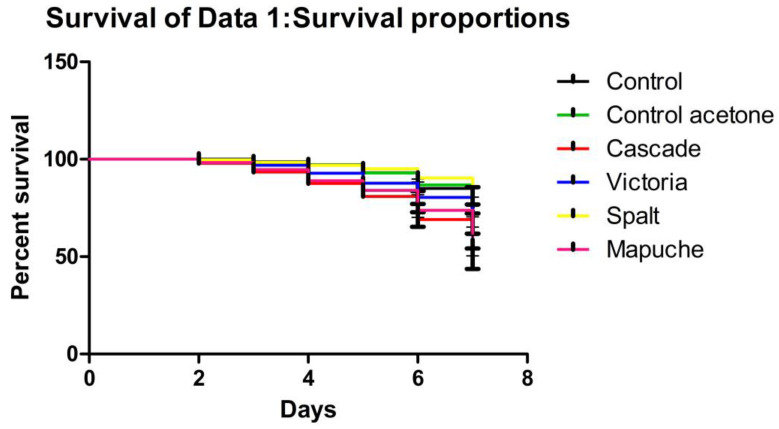
Kaplan—Meier plot of honey bee larvae survival function on different treatment: Control with no solvent (Control), control with solvent (Control acetone), hydrolats of Cascade variety (Cascade), hydrolats of Victoria variety (Victoria), hydrolats of Spalt variety (Spalt), and hydrolats of Mapuche variety (Mapuche). The varieties that differed from the control are Cascade (Log-rank (Mantel--Cox) Test *p* < 0.0001) and Mapuche (Log-rank (Mantel--Cox) Test *p* = 0.0015).

**Figure 2 plants-11-03329-f002:**
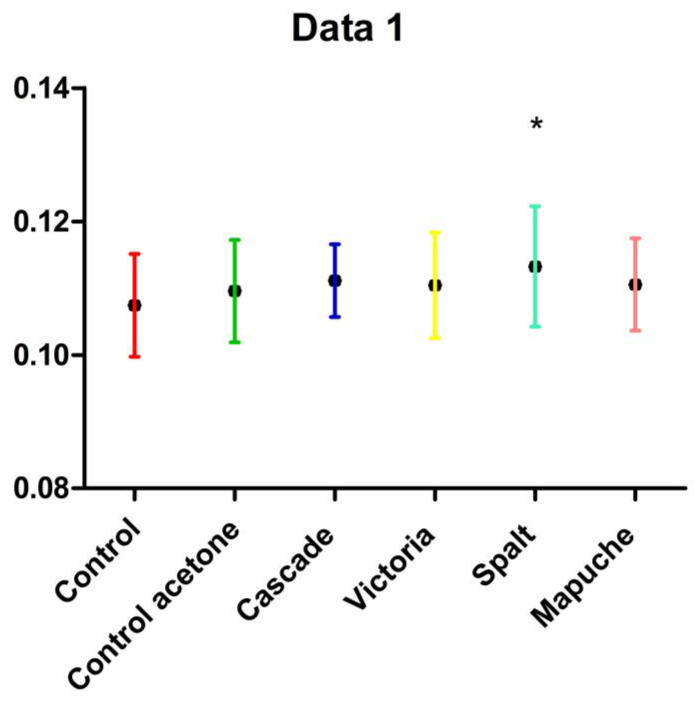
Means and SD of larval weights on day 7 with different treatments: Control with no solvent (Control), control with solvent (Control acetone), hydrolats of Cascade variety (Cascade), hydrolats of Victoria variety (Victoria), hydrolats of Spalt variety (Spalt), and hydrolats of Mapuche variety (Mapuche). “*” indicate statistical differences between treatments (*p* < 0.05).

**Table 1 plants-11-03329-t001:** Relative percentage compositions of hydrolats from *Humulus lupulus* female flower cones of Mapuche, Cascade, Spalt, and Victoria varieties.

Compound	KI (Exp)	KI (lit)	Mapuche	Victoria	Cascade	Spalt
				%	%	%
**Pentyl Acetate**	859	859			20.07	
**Beta-Linalool**	1088	1086	7.31	10.09	49.49	44.20
**Trans-Linalool Oxide**	1100	1102	1.51	1.95	5.79	
**NI**	1120	-	1.43	2.47	4.05	
**(+)-Limonene Oxide**	1134	1138	2.21	2.27		
**Isothujol**	1160	1157	1.32	1.44		
**(+)-Alpha-Terpineol**	1185	1189	7.24	1.27	4.51	
**Methyl 8-Nonynoate**	1199	1200		2.08		
**NI**	1277	-	2.05	2.87		
**Limonene diepoxyde**	1294	1294	6.49	6.25		
**9-oxadiciclo[3.3.1]nonane-2,7-diol**	1346	1347	1.62	1.13		
**Tetradecane**	1399	1399			17.43	
**(−)Beta-Caryophyllene**	1431	1430				18.77
**Alpha-Caryophyllene**	1469	1463				9.60
**Globulol**	1576	1576	5.94	9.74		
**Caryophyllene, Epoxide**	1597	1594	58.56	56.02	16.26	3.58
**NI**	1599	-				1.96
**Humuladienone**	1608	1607	2.44		2.45	
**Octadecano**	1800	1800				11.13
**2,6,10,14-tetrametilhexadecane**	1815	1815				10.75
**NI**	1860	-	1.88	2.42		

**Table 2 plants-11-03329-t002:** Content of saponins, polyphenols, flavonoids, and antioxidant capacity of the different hydrolats from *Humulus lupulus* female flower cones of Mapuche, Cascade, Spalt, and Victoria varieties.

Hydrolats	Saponins	Flavonoids	Polyphenols	Antioxidant Capacity
	AO µg/mL	Q µg/mL	AG µg/mL	TROLOX µg/mL
**Victoria**	648.7503	0.2507	133.2043	361.2587
**Cascade**	307.5998	0.0631	190.7868	217.3193
**Mapuche**	458.9114	0.0082	82.9924	157.8787
**Spalt**	129.9626	0.3261	210.7487	313.4731

**Table 3 plants-11-03329-t003:** Estimated LC_50_ (μL/mL) values for *Varroa destructor* obtained at each time interval for each hydrolats from *Humulus lupulus* female flower cones of Mapuche, Cascade, Spalt, and Victoria varieties.

Hydrolats	Mites LC_50_ (µL/mL)	
	24 h	48 h
**Cascade**	117.9 (47.6–292.0)	35.2 (19.6–63.0)
**Victoria**	16.1 (6.8–38.1)	1.3 (0.7–2.2)
**Spalt**	114.3 (25.9–503.7)	21.5 (7.6–60.7)
**Mapuche**	30.6 (9.5–98.5)	not estimated

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
