# Peer review of "Hydrolats from Humulus lupulus and Their Potential Activity as an Organic Control for Varroa destructor"

_plants, 2022, doi:10.3390/plants11233329_

Round 1

Reviewer 1 Report

This is a detailed manuscript regarding the appliance of hydrolats from Humulus lupulus as organic control against Varrosis. 

Notes:

- all Latin names must be written in italic

- the recommendation is to write V. destructor (as italic) in the whole manuscript; avoid writing just Varroa

- l 83 free space between 2020, 2021

- l 92 check spelling last author surname 

- l 101 find more appropriate word instead of little

- l 118 5 %

- l 141, 142 CHECK writing micrometer? USE mL

- L 148 move free spaces before Characterisation...

- l 152, 153 write abbreviations after first mentioning

- l 155, 156 check writing molecules

- l 162 move free spaces before From...

- l 173 420 mn

- l 221, 232 check manner of writing refs in text

- l 222 use as instead in 

- in vitro always as italic

- l 278 48 h

- l 280 move free spaces before The...

Figure 2. .... add mean +- SD 

- l 235 ...add refs doi.org/10.3390/app11188564

- l 326...add refs doi: 10.24099/vet.arhiv.0441

- l 341 move free spaces before According ...

- l 349 move free spaces before beta-...

- l 366 viability - move italic

- l 376, 384 move needless free spaces 

Authors' contributions - write according to Journal instructions

Author Response

Thanks to the reviewers' comments, we proceeded to correct the present work, which is attached below. 

Reviewer 2 Report

Varroa destructor is a parasitic mite, which is considered a severe pest for honey bees causing serious losses to the beekeeping. This study suggests the good in vitro efficacy of the Victoria hydrolat, presented a high acaricidal activity, null toxicity in adult bees, and a very low toxicity in larvae of honey bees. These results would indicate that Victoria hydrolat is a good compound to be used as an alternative for the control of V. destructor. The results of the paper are interesting, but the paper needs to be highly improved before publication in Plants. Some observations are listed below.

1. Materials and methods: the use of RIs in the identification of EO components should be inserted.

2. Table 1: insert a column with literature RIs; report also the % for chemical classes and the total identified.

3. The quality of Figure 1 needs to be improved.

4. Figure 1 and Figure 2 require difference analysis.

5. The format of references should be consistent, such as ref 15, full name or abbreviation of journal name.

Author Response

(The authors gave the same response as above.)

Reviewer 3 Report

Dear Authors,

The present paper, "Hydrolats from Humulus lupulus and their potential activity as an organic control for Varroa destructor", is partly innovative, showing the activity of the hydrolats of the hop. According to the data, the hydrolat, Victoria from studied hop varieties (Cascade, Spalt, Victoria and Mapuche) could meet the expectations of its use in beekeeping against Varroa destructor. The presented experiment and methodology in the study are well presented and planned to realize the aim of the study. Nevertheless, this applies to observations in laboratory conditions and not in an experimental apiary, even though the research material came from this apiary. This stage of the study only shows that isolated adult bees (five each) and Varroa showed 50% mortality as well as mortality of the L1-, L5-L6 larval stage, which is not generally hosts of Varroa destructor and the parasite does not reproduce on them. 

By exposing substances from the Victoria hop variety for a long time (24 and more hours), it can affect 1/13 of the developmental bee cells in which Varroa resides, i.e. on the 8th day of bee larval development, and the remaining most significant amounts of Varroa located under the wax seal (from 8 day), These substances, like many others, cannot affect this parasite due to the wax barrier. In the current study, a lethality index of 50% = LC50 was obtained for the tested substances, i.e. only 50% of 1/13 of the Varroa living in the larval form and 50% of the flying bee can be killed. The development of Varroa shows that for one development cycle lasting 13 days, the amount of this parasite in the hive at least doubles, which indicates that the use of this substance can kill only 50% and, at most, can stop the increase in the amount of Varroa in the bee colony. Still, it does not cause complete her cure. The effect of the substance at the level of 50% mortality is not satisfactory. Because the larval time of the bee under capping is 13 days, the hive should be exposed to the tested agent for at least that many days.

Nevertheless, a positive step in this research is demonstrating that these substances can be obtained as secondary products.

The presented study has a scientific context, so general information on the acaricidal effect of hop hydrolat should be published.

Notes to the text.

Lines: 304 not sure if this is an adult bee or a larve: ...Victoria presented a bee survival..... .

Fig 2. is weakly described ( the linear graph will be better).

Author Response

(The authors gave the same response as above.)

Round 2

Reviewer 2 Report

I think the modifications have improved the manuscript. But, please add manufacturer’s name and its address (city, country).

Author Response

Thank you for your comments, we have included the production data of the chemicals used. 
